# Interplay of Dietary Fatty Acids and Cholesterol Impacts Brain Mitochondria and Insulin Action

**DOI:** 10.3390/nu12051518

**Published:** 2020-05-23

**Authors:** Mareike Schell, Chantal Chudoba, Antoine Leboucher, Eugenia Alfine, Tanina Flore, Katrin Ritter, Katharina Weiper, Andreas Wernitz, Janin Henkel, André Kleinridders

**Affiliations:** 1Junior Research Group Central Regulation of Metabolism, German Institute of Human Nutrition, D-14558 Nuthetal, Germany; Mareike.Schell@dife.de (M.S.); Chantal.Chudoba@dife.de (C.C.); Antoine.Leboucher@dife.de (A.L.); Eugenia.Alfine@dife.de (E.A.); Tanina.Flore@dife.de (T.F.); Katrin.Ritter@dife.de (K.R.); weiper@uni-potsdam.de (K.W.); 2German Center for Diabetes Research (DZD), D-85764 München-Neuherberg, Germany; 3Department of Nutritional Biochemistry, Institute of Nutritional Science, University of Potsdam, D-14558 Nuthetal, Germany; jhenkel@uni-potsdam.de; 4Department of Molecular Epidemiology, German Institute of Human Nutrition, D-14558 Nuthetal, Germany; Andreas.Wernitz@dife.de; 5Department of Molecular and Experimental Nutritional Medicine, Institute of Nutritional Science, University of Potsdam, D-14558 Nuthetal, Germany

**Keywords:** cholesterol, insulin signaling, mitochondria, brain, inflammation, fatty acids, JNK, insulin receptor

## Abstract

Overconsumption of high-fat and cholesterol-containing diets is detrimental for metabolism and mitochondrial function, causes inflammatory responses and impairs insulin action in peripheral tissues. Dietary fatty acids can enter the brain to mediate the nutritional status, but also to influence neuronal homeostasis. Yet, it is unclear whether cholesterol-containing high-fat diets (HFDs) with different combinations of fatty acids exert metabolic stress and impact mitochondrial function in the brain. To investigate whether cholesterol in combination with different fatty acids impacts neuronal metabolism and mitochondrial function, C57BL/6J mice received different cholesterol-containing diets with either high concentrations of long-chain saturated fatty acids or soybean oil-derived poly-unsaturated fatty acids. In addition, CLU183 neurons were stimulated with combinations of palmitate, linoleic acid and cholesterol to assess their effects on metabolic stress, mitochondrial function and insulin action. The dietary interventions resulted in a molecular signature of metabolic stress in the hypothalamus with decreased expression of occludin and subunits of mitochondrial electron chain complexes, elevated protein carbonylation, as well as c-Jun N-terminal kinase (JNK) activation. Palmitate caused mitochondrial dysfunction, oxidative stress, insulin and insulin-like growth factor-1 (IGF-1) resistance, while cholesterol and linoleic acid did not cause functional alterations. Finally, we defined insulin receptor as a novel negative regulator of metabolically stress-induced JNK activation.

## 1. Introduction

The growing obesity pandemic is nowadays a global health concern and affects all age classes. Obesity is a major determinant for the establishment of insulin resistance and can lead to metabolic disorders, such as non-alcoholic fatty liver disease, type 2 diabetes and metabolic syndrome [1]. The development of obesity is mainly due to a lack of physical exercise with disproportional food intake. Feeding mice high caloric diets containing either high-fructose or high-fat concentrations induces obesity and insulin resistance [2]. In line with this, increased portion sizes with concomitant increased caloric intake cause obesity in humans [3]. Though high-fat diets (HFDs) have been shown to be instrumental for the induction of obesity and insulin resistance, it is of importance to differentiate between the quality of consumed fatty acids and their potential interplay with other nutrients. Not every high-fat diet has to exert only negative effects on metabolism. Feeding mice a HFD containing 45%–60% of calories derived mainly from lard is sufficient to induce obesity and insulin resistance. Yet, obese mice fed a HFD supplemented with the ω3 poly-unsaturated fatty acid, eicosapentaenoic acid, exhibit improved glucose tolerance and insulin sensitivity but did not decrease food intake [4]. These data indicate that the different HFDs exert diet-specific effects on insulin sensitivity, which also depends on genetic predispositions. In addition, feeding mice cholesterol-containing HFDs, containing either increased amounts of long-chain saturated fatty acids (LCSFA) or poly-unsaturated fatty acids (PUFA), induces body weight gain, fat mass accumulation and insulin resistance. Mice fed a cholesterol-containing low-fat diet developed hepatic steatosis, although they did not become obese [5].

The brain occupies a pivotal role in the regulation of body weight and insulin sensitivity. We have recently shown that feeding mice a HFD induces brain insulin resistance with a reduced mitochondrial stress response as early as three days of dietary exposure, while activation of brain insulin signaling counteracted these HFD-induced metabolic alterations [6]. This HFD contained high amounts of LCSFA, suggesting that an increased consumption of LCSFA is a potential mediator of brain insulin resistance. Indeed, the LCSFA palmitate is sufficient to induce hypothalamic insulin resistance [7,8], whereas long-chain mono-unsaturated fatty acids do not alter insulin sensitivity [9]. In addition, palmitate induces cell stress and activates cellular stress kinases, such as inhibitor of κB kinase, c-Jun N-terminal kinase (JNK) or protein kinase C. As these kinases have been linked to insulin resistance, it indicates that palmitate induces an inflammatory response with reduced insulin action in neurons [7,9,10]. Yet, humans do not consume exclusively only one fatty acid species but ingest a mixture of fatty acids, lipids, cholesterol and other nutrients, which modulate each other’s effects on metabolism. In line with this, it has been shown that oleate or, e.g., the PUFA docosahexaenoic acid, reverse palmitate-induced insulin resistance [11,12], highlighting the importance of understanding the interplay of fatty acids on cellular metabolism. An interplay of different fatty acids with cholesterol might also modulate insulin action. Cholesterol-containing diets have been shown to induce insulin resistance in peripheral tissues [13,14], but reduced cholesterol levels cause brain insulin resistance [15]. Although the brain is independent of dietary cholesterol, as it synthesizes its own cholesterol pool, high-fat/high-cholesterol intake has been shown to affect barrier integrity [16], which might cause altered neuronal homeostasis and metabolism [17]. Decreased mitochondrial cholesterol content deteriorates mitochondrial function [18]. In contrast, elevated cholesterol levels in mitochondrial membranes impair mitochondrial function [19] and mitochondrial cholesterol loading exacerbates inflammation [20]. These data indicate that increased as well as decreased cholesterol levels within mitochondria worsen their function, pointing to an important role of precise cholesterol regulation for brain health. Interestingly, insulin is a key regulator of brain cholesterol metabolism and reduced brain cholesterol synthesis is a consequence of impaired insulin sensitivity [21]. Conversely, reducing cholesterol levels in neurons induces insulin resistance [22], highlighting the interplay of cholesterol metabolism, insulin action and mitochondrial function. We have recently shown that mitochondrial dysfunction is cause and consequence of hypothalamic insulin resistance and can be induced by feeding mice a HFD [6,23]. It seems plausible that altered cholesterol levels with increased concentrations of LCSFAs might further deteriorate mitochondrial function and insulin signaling in the brain.

Increased dietary intake of PUFAs, especially ω3-PUFA, correlates with metabolic health [24], whereas an increased ω6- to ω3-PUFA ratio raises the risk for obesity [25]. We were able to show that a Western-type HFD containing soybean oil-derived PUFAs and cholesterol caused obesity with hepatic steatosis, massive liver inflammation, mitochondrial dysfunction and hepatic insulin resistance in mice, displaying many clinical parameters of patients with non-alcoholic steatohepatitis suffering from metabolic syndrome [5,26]. Interestingly, this diet was more detrimental for liver function than a combination of conventional HFD (containing mainly lard) with cholesterol. As soybean oil consists of high amounts of the ω6-PUFA linoleic acid, these data implicate that the interplay of ω6-PUFA with cholesterol is particularly harmful for metabolism, especially for liver function. Up to now, it is unclear how these diets affect brain metabolism, inflammation and mitochondrial function.

In this study, we investigated the effect of a standard chow diet (STD), 0.75% cholesterol in a standard diet (CHO + STD), 0.75% cholesterol in a HFD containing ω6-PUFA-rich soybean oil (CHO + SOY), 0.75% cholesterol in a HFD containing mainly lard as a fat source (CHO + LAR) or a HFD (containing mainly lard as a fat source, LAR) on hypothalamic stress responses and homeostasis in mice. Further, we tested whether cholesterol, the LCSFA palmitate or ω6-PUFA linoleic acid, or a combination of these fatty acids with cholesterol, impairs mitochondrial function and insulin action in hypothalamic neurons.

All tested dietary interventions cause a molecular signature of metabolic stress in the hypothalamus with decreased markers of blood–brain barrier integrity, mitochondrial function, elevated protein carbonylation and JNK activation. Palmitate causes mitochondrial dysfunction, oxidative stress and insulin as well as insulin-like growth factor-1 (IGF-1) resistance in vitro, while cholesterol and linoleic acid do not cause functional alterations. Overall, using in vivo and in vitro models, we (A) detect palmitate as a negative regulator of hypothalamic insulin receptor (IR) and insulin-like growth factor-1 receptor (IGF-1R) signaling, as well as of mitochondrial function, (B) reveal that cholesterol and ω6-PUFA treatment of hypothalamic neurons is not detrimental for insulin action or mitochondrial activity and (C) identify IR as a novel modulator of palmitate-induced JNK activation.

## 2. Materials and Methods

All chemicals were of analytical or higher grade and obtained from local providers, unless otherwise stated.

### 2.1. Animals and Experimental Design

Male C57BL/6JRj mice were group-housed in type II cages at 20 ± 2 °C with a 12 h light/dark-cycle and fed a standard chow diet (STD), 0.75% cholesterol in a standard diet (CHO + STD), 0.75% cholesterol in a high-fat diet containing ω6-PUFA-rich soybean oil (CHO + SOY), 0.75% cholesterol in a high-fat diet containing mainly lard as a fat source (CHO + LAR) or a high-fat diet containing mainly lard without additional cholesterol (LAR) for 20 weeks. Detailed diet composition is shown in Appendix A, as previously described [5]. Animal experiments were performed according to the ARRIVE guidelines. Treatment of the animals followed the German animal protection laws and was performed with the approval of the state animal welfare committee (LAVG, Brandenburg). The study was conducted in accordance with the Declaration of Helsinki, and the protocol was approved by the Ethics Committee of the state of Brandenburg (TVA 2347-18-2013).

### 2.2. In Vitro Stimulations

For all stimulations, immortalized hypothalamic CLU183 (mHypoA-2/23 CLU183) cells were cultivated in Dulbecco’s Modified Eagle’s Medium (DMEM) GlutaMAX high-glucose (Gibco), supplemented with 1 mM sodium pyruvate (Gibco), 10% fetal bovine serum (Pan, South Africa) and 1% penicillin-streptomycin (Gibco). CLU183 insulin receptor knockout (IR KO) cells were generated as previously described [6]. All cell cultures were maintained at 37 °C with 5% CO_2_. For all experiments, cells were seeded one day before the stimulation. CLU183 cells were incubated for 16 h with 5 µM cholesterol (complexed with 50 µM methyl-β-cyclodextrin (MβCD)), 250 µM linoleic acid or palmitic acid (LA or PA, both hydrolyzed under alkaline conditions and coupled to bovine serum albumin (BSA), as described previously [27]), or respective controls (5 µM MβCD and/or 125 µM BSA). The used concentrations were consistent through all in vitro experiments.

For insulin stimulation, CLU183 cells were first stimulated with cholesterol and/or fatty acids for 16 h, and were then serum-deprived for 3 h with DMEM GlutaMAX high-glucose, 1 mM sodium pyruvate and 1% penicillin-streptomycin, and subsequently stimulated with 100 nM insulin (Sigma-Aldrich, Taufkirchen, Germany) for 5 min.

For inhibition of the IR, CLU183 cells were first stimulated with palmitate with and without cholesterol for 16 h, were then serum-deprived for 3 h in the presence of 100 nM IR antagonist S961 (Novo Nordisk), and finally stimulated with 100 nM insulin (Sigma-Aldrich, Taufkirchen, Germany) for 5 min.

For inhibition of JNK, CLU183 cells were stimulated with 10 µM JNK-inhibitor SP600125 (Sigma-Aldrich, Taufkirchen, Germany) and with cholesterol and/or fatty acids for 16 h, and were then serum-deprived for 3 h in the presence of SP600125 and additionally stimulated with 100 nM insulin for 5 min. Dimethyl sulfoxide (Sigma-Aldrich, Taufkirchen, Germany) was used as a solvent control for SP600125 (= –SP600125).

### 2.3. Ex Vivo Stimulations

Cholesterol and fatty acid stimulation experiments were performed on coronal slices of eight 19–21 weeks old male C57BL/6N mice, which were killed by cervical dislocation. After carefully removing the brain from the skull, the brain was placed into a brain matrix (Zivic Instruments, Pittsburgh, PA, United States) to isolate the hypothalamus (bregma -1.34 mm to bregma -1.74 mm). Consecutive coronal slices of 300 µm were cut using a vibration microtome (Leica, Wetzlar, Germany), and were then placed in artificial cerebrospinal fluid [28] and oxygenized with 5% CO_2_/95% O_2_. Following 1 h recovery, slices were stimulated for 5 h with either 5 µM cholesterol, 250 µM LA, 250 µM PA, or a combination of cholesterol and fatty acids, as well as their respective control BSA with MβCD, and were subsequently stimulated with 100 nM insulin for 15 min. The protocol was approved by the Ethics Committee of the state of Brandenburg (T-07-19-CRM).

### 2.4. Serum Analysis

Insulin levels were measured using an insulin (ELISA) kit (Crystal Chem; Downers Grove, IL, United States). Analysis of fatty acid spectra of serum phospholipids (PL) was performed with a strongly modified method using extraction with tert-butyl methyl ether/methanol, solid-phase separation, hydrolysis and methylation with trimethyl sulfonium hydroxide, and subsequent analysis by gas chromatography [29,30,31] and a flame ionization detector. Modifications of the analysis method were previously published [32]. In this study, 50 µL serum samples were processed as described and were then subjected to a bonded phase column separation after redissolving the dried lipids in chloroform. Fatty acid composition of serum PL was expressed as area percentage of each fatty acid relative to total area of all detected fatty acids: C12:0, C14:0, C15:0, C16:0, C16:1n7c, C17:0, C18:0, C18:1n9c, C18:1n7c, C18:2n6c, C20:0, C18:3n6, C18:3n3, C20:1n9, C20:2n6, C20:3n6, C20:4n6, C20:5n3, C22:4n6, C22:5n6, C22:5n3, C22:6n3.

### 2.5. Cholesterol Assay

Total and free cholesterol in hypothalamic tissue and cell culture experiments were determined as described previously [5], with minor modifications. Briefly, frozen tissue or cell homogenates homogenized in lysis buffer using a pestile mixer or sonicator were heated, centrifuged and supernatants were subsequently incubated with an assay buffer containing 100 mmol/L phosphate buffer (pH 7.4), 0.026% Triton X-100, 1 mmol/L sodium cholate, 0.63 mg/mL p-hydroxyphenylacetic acid, 0.5 U/mL cholesterol oxidase and 0.2 U/mL peroxidase without or with 0.5 U/mL cholesterol esterase. Fluorescence was detected after 40 min incubation at 37 °C with 325 nm (excitation) and 415 nm (emission). The esterified cholesterol was quantified by the difference between total and free cholesterol.

### 2.6. Western Blot and Protein Carbonylation Assay

Western blot analysis was performed as described previously [5,6] using anti-occludin (NBP1-87402) obtained from Novusbio, Total OXPHOS Rodent WB Antibody Cocktail (ab110413) and anti-PGC-1α (ab54481) obtained from Abcam (Cambridge, UK), anti-HSP60 (sc-376240) obtained from Santa Cruz and anti-phospho-SAPK/JNK (Thr183/Tyr185) (#9251), anti-JNK2 (#9258), anti-SOD2 XP (#13141), anti-SIRT3 (#5490), anti-phospho-AKT (Ser473) (#9271), anti-AKT (#9272), anti-IRβ (#3025) and anti-IGF-1Rβ (#3027) antibodies, as well as the secondary antibodies anti-rabbit antibody (#7074) and anti-mouse antibody (#7076) obtained from Cell Signaling (Cambridge, UK). Ponceau staining served as a loading control. Oxyblot analysis was carried out as previously published [33] with anti-DNP antibody after membrane derivatization (D9656, Sigma-Aldrich, Taufkirchen, Germany). Specific bands were detected by using a chemiluminescence reagent in the ChemiDoc™ Imaging System with ImageLab software (Bio-Rad, Munich, Germany). Band intensities were quantified via densitometric analysis using Image Lab 5.2.1 and Image J software and were normalized to protein content exemplified by Ponceau staining or total unphosphorylated proteins (JNK and AKT phosphorylation).

### 2.7. Gene Expression Analysis

Total RNA was extracted from 3–4 × 10^5^ cells with QIAzol Lysis Reagent (Qiagen, Hilden, Germany) or RNeasy Kit (Qiagen). Overall, 1 μg of RNA from cells was reverse transcribed in 20 μL using Random hexamer primers (11034731001, Hoffmann-La Roche, Basel, Switzerland), Thermo Scientific™ dNTP-Set, and M-MLV Reverse Transcriptase (Promega GmbH, Walldorf, Germany). Real-time PCR was performed and analyzed as previously published [6] and primer sequences are listed in Appendix A. Gene expression was calculated according to the ΔΔCT method using Tbp (TATA-box binding protein) or β-Actin as a reference gene. The specificity of SYBR Green primers was confirmed by melting curve analysis.

### 2.8. Genomic DNA Isolation

DNA from cells was extracted using the Invisorb Spin Tissue Mini Kit (Invitek Molecular GmbH, Berlin, Germany) following the manufacturer’s manual.

### 2.9. Mitochondrial Respiration

Differences in mitochondrial respiration were determined using the Seahorse XF Mito Stress Test Kit and the Seahorse XF96 extracellular Bioflux analyzer (Agilent, Santa Clara, CA, United States), measuring oxygen consumption rate (OCR) and extracellular acidification rate (ECAR) of adherent cells to test mitochondrial function. All compound concentrations were tested and optimized before the assay and were consistent through all Seahorse runs for all experimental setups [6]. Final concentrations for the compounds were 2 μM for Oligomycin (Port A), 0.5 μM for carbonyl cyanide-4-(trifluoromethoxy)phenylhydrazone (FCCP) (Port B) and 1 μM for Rotenone/2 μM for Antimycin A (Port C). Cells were seeded at a density of 5000 cells/well two days prior to the experiment within a 96-well microplate. Four wells were prepared without cells as background signal (‘blank’). On the following day, cells were stimulated with cholesterol and/or fatty acids as well as the respective controls for 16 h overnight. For IR inhibition, cells were stimulated with palmitate with or without cholesterol for 13 h and with 100 nM S961 for an additional 3 h. Prior to the assay, cells were washed once with Seahorse Medium (XF base Minimum DMEM supplemented with 1 mM sodium pyruvate, 2 mM glutamine and 10 mM glucose, pH 7.4) and then incubated in the final amount of assay medium in a non-CO_2_ incubator at 37 °C to maintain pH levels. After calibration, the microplate was placed in the Seahorse Bioflux analyzer and the experiment was performed according to manufacturer’s instructions (3 min mix, 3 min measure; 3 cycles for each port). OCR data were normalized to protein content and were analyzed using Wave 2.4.0 software (Agilent, Santa Clara, CA, USA).

### 2.10. Statistical Analysis

Two groups were compared using the unpaired two-tailed Student’s *t*-test or the Mann–Whitney test when necessary. The statistical significance between differences of more than two groups was determined by one-way analysis of variance (ANOVA) or two-way ANOVA with Tukey’s post hoc test for multiple comparisons, or the Kruskal–Wallis test for non-parametric samples, as detailed in the legends to the figures using GraphPad Prism version 8 for Windows (GraphPad Software, La Jolla, CA, USA). Differences with a *p*-value ≤ 0.05 were considered statistically significant.

## 3. Results

### 3.1. Cholesterol/PUFA Diet Increases Both Cholesterol and Metabolic Stress in the Brain

Mice were divided into five groups and were fed different diets with altered fat composition. For this purpose, 6-week-old mice were either fed a STD, CHO + STD, CHO + SOY (contains high amounts of soybean oil-derived PUFAs), CHO + LAR (contains high amounts of LCSFAs) or LAR for 20 weeks, as reported previously [5]. Animals on all high-fat diets gained more weight than animals fed either a chow diet or cholesterol-enriched chow diet (already published in Reference [5] and Figure 1A). Despite similar weight gain between mice fed a CHO + SOY, CHO + LAR or LAR diet, animals fed the LAR diet were significantly more insulin resistant than other groups, as evidenced by four-fold increased fasting serum insulin levels and higher blood glucose levels in response to glucose or insulin administration compared to STD control, confirming the detrimental effect of LCSFA-containing diets on insulin sensitivity (Figure 1B, Appendix A).

As the CHO + SOY compared to CHO + STD and CHO + LAR diet groups had an enormous effect on liver function and caused massive hepatic inflammation [5], we assessed their effects on brain homeostasis and mitochondrial dysfunction. To exemplarily validate the SOY feeding regime, we determined the relative abundance of ω6-PUFA linoleic acid (LA) in serum of mice fed a CHO + SOY diet compared to the STD diet group. Indeed, feeding mice a CHO + SOY diet increased LA abundance by ~37% compared to STD control, identifying a successful enrichment of LA in the serum in CHO + SOY-fed mice (Appendix A). Overall, the CHO + SOY diet caused a general altered profile of fatty acids in the serum, showing the extensive impact of the diet on serum fatty acid abundance (Appendix A).

We then assessed whether cholesterol-supplemented diets were able to increase cholesterol content in the hypothalamus. This analysis revealed that, unexpectedly, only the CHO + SOY diet, but not CHO + STD or CHO + LAR diets, increased cholesterol levels in the hypothalamus, with a 61% increase of esterified cholesterol and about a 20% increase of free and total cholesterol compared to the STD group (Figure 1C). The majority of dietary cholesterol is not able to penetrate the blood–brain barrier in healthy conditions [34] and thus, can only enter the brain via a disruption of the blood–brain barrier (BBB). Interestingly, occludin protein expression, which is a marker for BBB integrity, was decreased in the hypothalamus of all mice fed a cholesterol-containing diet compared to STD control (Figure 1D). Yet, cholesterol levels were only increased in the CHO + SOY group (Figure 1C), suggesting that an interaction of soybean oil-derived PUFAs with cholesterol is responsible for the elevated cholesterol levels in the hypothalamus. As the deterioration of the BBB can harm the brain, we further investigated the activation of the serine/threonine stress kinase JNK in hypothalamic samples of the different mouse groups. Only mice fed a LAR, CHO + LAR or CHO + SOY diet caused increased JNK activation with elevated Thr183/Tyr185 phosphorylation, while a CHO + STD diet did not affect JNK activation. In detail, LAR-fed mice exhibited a 3.7-fold increase in JNK activation, whereas CHO + LAR- and CHO + SOY-fed mice showed a ~2.3-fold increase compared to STD control. Interestingly, p46 JNK was more strongly activated by LAR, CHO + LAR and CHO + SOY than p54, which was barely detectable (Figure 1E).

### 3.2. Specific Alterations of Mitochondrial Protein Expression Due to Cholesterol-Containing Diets

As elevated concentrations of both LCSFAs and ω6-PUFAs correlate with metabolic alterations and impaired insulin action in obesity, we further investigated hypothalamic mitochondrial protein homeostasis, which is under the control of insulin [6]. First, we investigated the expression pattern of subunits of the electron transport chain complexes I, II, III, IV and V (ATP synthase), and the mitochondrial antioxidative enzyme superoxide dismutase 2 (SOD2). This analysis revealed that mice fed either LAR- or cholesterol-containing diets showed reduced protein expression of subunit ATP5A (ATP synthase subunit alpha) of complex V (Figure 2A,B). A similar phenotype was also observed for complex III regulation with decreased expression of its subunit UQCRC2 (Cytochrome b-c1 complex subunit 2), while the LAR diet did not affect its expression. Interestingly, protein expression of NDUFB8 (NADH:Ubiquinone Oxidoreductase Subunit B8) and SDHB (succinate dehydrogenase complex subunits B), subunits of complex I and II, were increased in the hypothalamus of mice fed a LAR diet, indicating that each diet causes a unique alteration of the hypothalamic mitochondrial proteome (Figure 2A,B). In contrast, protein levels of the mitochondrial antioxidative enzyme SOD2 were indistinguishable between all tested groups (Figure 2A,B).

Next, we assessed protein levels of regulators of mitochondrial function, the master regulator of mitochondrial biogenesis peroxisome proliferator-activated receptor-gamma coactivator 1α (PGC1α), the main mitochondrial matrix chaperone heat-shock protein 60 (HSP60), as well as the mitochondrial deacetylase sirtuin 3 (SIRT3)—all proteins whose dysregulations affect insulin action [23,35,36]. PGC1α was reduced in all tested groups, whereas HSP60 or SIRT3 were only reduced in mice fed a LAR or CHO + LAR diet, confirming that different HFDs induce distinct alterations of mitochondrial protein expression (Figure 2C,D). As alterations in the mitochondrial proteome can cause cellular stress [37], we further assessed protein carbonylation—a marker of oxidative stress—in the hypothalamus of these mice. Total protein carbonylation was only slightly altered in the hypothalamus, yet with a significant increase in mice fed a LAR diet when comparing to STD, and surprisingly, a minor decrease in CHO + SOY-fed mice compared to STD and CHO + STD control (Figure 2E,F). Furthermore, feeding mice a CHO + LAR, as well as LAR diet, caused increased protein carbonylation compared to the CHO + STD and CHO + SOY diets. As the majority of carbonylated proteins seemed to be larger than 150 kDa, we additionally only analyzed proteins with high molecular weight, confirming our previous observation that both the CHO + LAR and LAR diets show an increase in carbonylated proteins compared to CHO + SOY (Figure 2E,F). Overall, this analysis revealed that each feeding regime elicits distinct alterations in mitochondrial protein expression with reduced expression of subunits of complex III and V in cholesterol-containing diets and slightly elevated protein carbonylation, especially in the hypothalamus of mice fed LAR-containing diets.

### 3.3. Palmitate but not Cholesterol or Linoleate Decreases Mitochondrial Function

To gain detailed insights into metabolic effects of fatty acids enriched in these diets cholesterol, or the combination of cholesterol with these fatty acids, we treated the hypothalamic cell line CLU183 with 5 µM cholesterol, 250 µM of palmitic acid (PA) and linoleic acid (LA), or a combination of cholesterol with these fatty acids, for 16 h and assessed mitochondrial function. 5 µM of cholesterol was chosen, as this lowest concentration did not decrease cell viability (Appendix A). We used methyl-β-cyclodextrin (MβCD) to complex cholesterol and achieve cholesterol uptake in vitro. As MβCD per se reduces cholesterol levels in neurons [22], we used only 5 µM MβCD as solvent control for cholesterol-treated cells to avoid artificial cholesterol depletion in our control conditions [38,39]. In comparison with 50 µM, 5 µM MβCD did not alter gene expression of the cholesterol biosynthesis pathway or insulin sensitivity (data not shown). LA and PA were coupled to bovine serum albumin (BSA) and thus, we used BSA as a control for these conditions.

To confirm successful cholesterol treatment, we analyzed cholesterol accumulation and cholesterol-regulated gene expression in cholesterol-treated CLU183 cells. In line with our in vivo data, this analysis revealed that only the combination of LA with cholesterol was sufficient to cause a significant ~30% and 41% increase of free and total cholesterol levels in neurons, while esterified cholesterol levels were unchanged (Figure 3A). As elevated cholesterol concentrations are able to inhibit endogenous cholesterol biosynthesis, we assessed gene expression levels of the cholesterol biosynthesis pathway. This analysis revealed reduced gene expression of sterol regulatory element-binding protein 2 (*Srebp2*), farnesyl diphosphate synthase (*Fdps*) and squalene epoxidase (*Sqle*) in cholesterol-treated cells, whereas 3-hydroxy-3-methylglutaryl-CoA reductase (*Hmgcr*), the rate-controlling enzyme of the mevalonate pathway, was not affected (Appendix A). These data indicate that CLU183 cells were able to take up and metabolize extracellular cholesterol, as evidenced by increased intracellular cholesterol accumulation and reduced expression levels of cholesterol-regulated genes. Following this, we determined mitochondrial function using a Seahorse Bioflux analyzer. Interestingly, only PA and the CHO + PA treatment reduced basal respiration by 65% (PA) and 52% (CHO + PA), reduced maximal respiration by 59% and 55% respectively, with an additional 52% and 44% reduction in ATP production (Figure 3B) and with overall reduced energy metabolism, as both PA and CHO + PA also decreased extracellular acidification rate (ECAR) (data not shown). Cholesterol treatment did not affect mitochondrial function nor did it change the combination of cholesterol with PA or LA effect of these fatty acids on mitochondrial function (Figure 3B).

To understand why PA caused mitochondrial dysfunction, we performed a detailed mitochondrial analysis of subunits of the electron transport chain complexes, SOD2, as well as gene expression of *Pgc1α*, *Hsp60* and *Sirt3*, similar to our in vivo study. This analysis revealed that neither cholesterol, LA, PA, nor a combination affected protein expression of the electron transport chain complexes along with unaltered SOD2 protein expression (Figure 3C,D). In addition, mitochondrial DNA content was unaltered, indicating that PA does not affect mitochondrial function by decreasing mitochondrial mass (Appendix A). In line with this, neuronal *Hsp60* mRNA levels were unaffected by different treatments, while *Sirt3* and *Pgc1α* gene expression were significantly reduced by cholesterol treatment (Appendix A). As cholesterol did not change mitochondrial function, the reduction of *Sirt3* and *Pgc1α* mRNA levels in any CHO-treated cells, including CHO + PA treatment, could not account for the observed decreased mitochondrial function in CLU183 cells treated with PA. Next, we assessed markers of mitochondrial dynamics. While PA treatment did not change the gene expression of *Mfn1* (*Mitofusin-1*) or *Drp1* (*Dynamin-related protein 1*) (Appendix A), PA treatment decreased *Opa1* (*OPA1 mitochondrial dynamin-like GTPase*) expression (Appendix A), suggesting that PA treatment induces mitochondrial fission along with reduced mitochondrial activity. Surprisingly, CHO + PA reduced mitochondrial respiration but did not affect mitochondrial dynamics.

Based on these findings, we investigated oxidative stress by assessing protein carbonylation. Similar to our observed protein carbonylation results in vivo, PA, but also both PA and LA with cholesterol treatment, caused increased total protein carbonylation as well as carbonylated proteins at high molecular weight (>150 kDa) in CLU183 neurons compared to control-treated neurons (Figure 3E,F). In summary, PA causes oxidative stress and mitochondrial dysfunction, while cholesterol does not affect mitochondrial activity in these experimental setups.

### 3.4. Palmitate, but not Cholesterol or Linoleic Acid, Induces Insulin and IGF-1 Resistance with Increased Inflammation in Hypothalamic Neurons

We have previously shown that dietary intake of soybean oil-derived PUFAs with cholesterol caused hepatic inflammation and insulin resistance [5]. In addition, our in vitro data show that palmitate treatment causes mitochondrial dysfunction in hypothalamic neurons, a phenomenon that can cause insulin resistance [23]. By performing in vitro insulin stimulation experiments, we tested whether PA or LA affected insulin sensitivity. As IR and IGF-1R signaling is difficult to distinguish [40], we investigated the effects of fatty acids in control and CLU183 cells deficient for the insulin receptor (IR KO) (Appendix A). To activate IR and IGF-1R, we used 100 nM insulin, which is sufficient to potently cross-activate the IGF-1 receptor [40]. Thus, we treated control and IR KO CLU183 cells with PA, LA or BSA as control, followed by insulin stimulation. These experiments showed that PA, but not LA, caused insulin resistance in control cells, as evidenced by a ~49% reduction in Ser473 phosphorylation of AKT (Figure 4A). Unexpectedly, control and IR KO cells exhibited the same degree of insulin-induced AKT phosphorylation under all tested conditions, suggesting a compensatory mechanism in IR KO cells. Indeed, IR KO cells exhibited an almost 7-fold increase in IGF-1R gene and protein expression compared to control, explaining the lack of reduced insulin-induced AKT activation in these cells (Appendix A). Interestingly, IR KO cells exhibited a similar reduction of insulin-induced AKT phosphorylation after PA treatment, showing that palmitate also reduces insulin-induced IGF-1R activation or causes IGF-1 resistance (Figure 4A).

We further tested the effect of PA, LA, cholesterol and their combination on mitochondrial function in IR KO cells using the Seahorse Bioflux analyzer. This analysis revealed that IR KO cells exhibited increased basal respiration compared to control cells, which was presumably due to elevated IGF-1R expression. Yet, PA or CHO + PA treatment caused a similar reduced basal respiration with a stronger suppression of mitochondrial activity in IR KO cells (Appendix A), confirming that IR expression is a crucial modulator of hypothalamic mitochondrial function [6] and suggesting that IR is vital to counteract the negative effects of PA on cellular homeostasis. Furthermore, neuronal *Pgc1α* and *Hsp60* mRNA levels were unaffected, while gene expression of *Mfn1*, *Opa1* and *Drp1* was significantly increased in IR KO cells (Appendix A), pointing to increased mitochondrial dynamics.

Overconsumption of HFD with high amounts of palmitate causes neuroinflammation. We identified increased activation of the stress kinase JNK in our in vitro settings (Figure 4B, Appendix A). We were able to identify a 5.4-fold increase of palmitate-induced p54 JNK Thr183/Tyr185 phosphorylation in control cells, which was not altered by short-term insulin stimulation (Figure 4B). Yet, comparing JNK activation between control and IR KO cells revealed a significant almost 2-fold enhanced palmitate-induced JNK activation in IR KO cells, indicating that the presence of IR is anti-inflammatory and cannot be compensated by endogenous IGF-1R overexpression (Figure 4B).

Next, we investigated whether cholesterol or a combination of cholesterol with PA or LA affected neuronal insulin sensitivity. To enable the comparison of all tested combinations, we examined the effect of used control substances (BSA for PA or LA, MβCD for cholesterol, BSA and MβCD for cholesterol with PA or LA) on insulin sensitivity, showing that controls exhibited similar insulin sensitivity (Appendix A). Subsequently, control and IR KO CLU183 cells were treated with cholesterol and fatty acids, followed by insulin stimulation. CHO treatment did not alter insulin sensitivity, but CHO + LA increased insulin sensitivity in control and IR KO cells compared to control-treated cells, whereas CHO + PA reduced insulin sensitivity compared to CHO + LA-treated cells, as evidenced by a 1.7-fold increase and a ~50% decrease in Ser473 phosphorylation of AKT, respectively (Figure 4C). In line with the effect of PA on JNK activation, we only identified increased JNK activation in CHO + PA-treated neurons (Figure 4D, Appendix A) with a 3.9-fold increase of p54 JNK Thr183/Tyr185 phosphorylation. Additionally, these data confirmed the protective effect of IR on aberrant JNK activation, as IR KO cells revealed elevated CHO + PA-induced JNK phosphorylation compared to control, with an overall significant genotype effect (Figure 4D). Interestingly, CHO + LA treatment enhanced insulin sensitivity, while LA treatment did not alter insulin action.

To investigate whether CHO + PA compared to PA treatment had an additive effect on insulin resistance and JNK activation, we directly compared PA with the CHO + PA treatment. These experiments revealed that the addition of cholesterol had no further effect on the PA-induced insulin resistance with similar Ser473 AKT- and Thr183/Tyr185 JNK-phosphorylation (Figure 4E,F, Appendix A).

### 3.5. Palmitate- and Palmitate/Cholesterol-Induced Insulin Resistance is Independent of JNK Activation

Next, we tested whether the PA- and CHO + PA-induced JNK activation was responsible for the observed insulin resistance phenotype in CLU183 cells using JNK inhibitor SP600125. While SP600125 successfully inhibited PA-induced JNK activation in both control and IR KO cells, JNK inhibition was not able to reverse PA- or CHO + PA-induced reduction of insulin-stimulated AKT-phosphorylation in both cell lines (Appendix A). Interestingly, JNK inhibition was more potent in control cells compared to IR KO cells, confirming that decreased IR signaling results in enhanced palmitate-induced JNK activation and cellular stress.

To further investigate a potential, differential effect of PA and CHO + PA on IGF-1R signaling in hypothalamic neurons, we inhibited IR action using the high-affinity IR peptide antagonist S961 at a concentration of 100 nM [41,42]. This analysis revealed that both PA and CHO + PA reduced insulin-induced Ser473 AKT phosphorylation, which was aggravated by the inhibition of the IR (Figure 5A). These data clearly show that PA and CHO + PA potently reduce IR and IGF-1R signaling and that the observed similar insulin sensitivity of control and IR KO cells was presumably based on the compensatory upregulation of IGF-1R (Figure 4C, Appendix A).

To investigate the effect of IR inhibition on mitochondrial function in cells without compensatory elevated IGF-1R expression, we treated cells with PA or CHO + PA and added S961. As shown earlier (Figure 3B), PA and CHO + PA caused a decrease in mitochondrial respiration. Surprisingly, S961 significantly reduced basal respiration only in control-treated cells and cells treated with PA but not with CHO + PA (Figure 5B), suggesting that proper IR action is important for mitochondrial activity.

### 3.6. Only Palmitate Induces Insulin Resistance on Hypothalamic Brain Slices

In the brain, there is a functional and metabolic interplay of neurons with astrocytes and microglia, which might cause different sensitivities to fatty acids and cholesterol-induced effects on insulin signaling. To assess this interplay in the hypothalamus, we cultivated coronal brain slices in oxygenated, artificial cerebrospinal fluid and added cholesterol, PA, LA or the combination of fatty acids with cholesterol to this medium. This treatment was followed by a 15 min insulin stimulation and Western blot analysis of Ser473 AKT phosphorylation. This experimental setup should reveal whether the interplay of different cell populations might modulate the effect of tested fatty acids and cholesterol on insulin action. This analysis showed that PA treatment was sufficient to induce brain insulin resistance with a reduction in insulin-induced Ser473 phosphorylation of AKT by 43%, while, unexpectedly, CHO + PA treatment did not result in a markedly decreased phosphorylation of AKT (Figure 6A,B). These data confirm the detrimental effect of palmitate on hypothalamic insulin signaling, while the CHO + PA treatment did not cause a significant decrease in insulin-induced Ser473 AKT phosphorylation, indicating that cholesterol treatment differentially regulates insulin action on brain slices compared to neuronal stimulations in vitro.

## 4. Discussion

This study investigated the effect of different fatty acids and cholesterol on metabolic stress, mitochondrial function and insulin signaling in hypothalamic neurons. Our data revealed that the investigated cholesterol-containing diets (CHO + STD and CHO + LAR), as well as the conventional high-fat diet (LAR) increased oxidative stress in the brain with slightly increased protein carbonylation. In addition, all HFD treatments reduced protein expression of PGC1α and HSP60 (Figure 2). Further, only cholesterol-containing diets reduced the expression of occludin (Figure 1D), a marker of BBB integrity. As the activation of the stress kinase JNK was only enhanced in hypothalamic samples of mice fed a LAR, CHO + LAR or a CHO + SOY diet, but not a CHO + STD diet, these data indicate that overall, increased inflammation is not the causal factor for the reduction of occludin (Figure 1E) and other yet undefined mechanisms cause an altered BBB integrity [43].

Treating neurons with PA, the most frequent saturated fatty acid in lard, or the combination of PA and cholesterol, intensified JNK activation and caused insulin resistance in vitro (Figure 4E,F). Interestingly, PA-induced insulin resistance was independent of JNK-activation as the inhibition of JNK did not reverse insulin resistance. The present study further showed that cholesterol might even be protective against palmitate-induced hypothalamic insulin resistance ex vivo, as CHO + PA was not as detrimental as PA treatment to insulin action on hypothalamic brain slices (Figure 6A,B). But, cholesterol treatment per se did not alter mitochondrial function, insulin action and palmitate-induced insulin resistance in vitro. Similarly, LA and a combination of LA and cholesterol were not detrimental for mitochondrial function or insulin signaling. Importantly, we demonstrated that the presence of IR is a crucial negative modulator of PA-induced JNK activation. This suggests that one important aspect of brain insulin signaling is to fine-tune brain function and ensure a balanced metabolism.

### 4.1. Impact of Cholesterol on Brain Insulin Resistance and Mitochondrial Function

In contrast to peripheral cholesterol metabolism, nearly all brain cholesterol is independently regulated and synthesized by de novo synthesis [44]. The cholesterol movement in and out of the central nervous system is controversially discussed. Due to the lack of direct experimental evidence, it is expected to be very unlikely that cholesterol-containing lipoproteins cross the BBB (Reference [45] and references therein). We showed that only the combination of CHO + SOY elevates cholesterol levels in the hypothalamus (Figure 1C), whereas all cholesterol-containing diets led to decreased protein expression of occludin (Figure 1D), indicating that prolonged excessive dietary intake of cholesterol might harm brain physiology by reducing BBB integrity [46]. Still, it is unclear whether this reduction is due to the presence of increased cholesterol concentrations, and it might be due to overall increased inflammation or other yet undefined mechanisms that might cause an altered BBB integrity [43]. Interestingly, only neurons treated with LA and cholesterol exhibited increased cholesterol levels (Figure 3A), but cholesterol treatment in general decreased expression levels of genes involved in endogenous cholesterol biosynthesis (Appendix A).

Cholesterol is an essential structural component for cell membranes and modulates membrane fluidity. While the central nervous system accounts for only 2%–3% of the whole body mass, it contains ~25% of the whole body’s cholesterol [44]. Cholesterol homeostasis is strictly regulated. Deficiency as well as aberrant storage of brain cholesterol can have profound consequences on neuronal survival, causing neurodegeneration [47]. Brain insulin resistance associates with neurodegenerative diseases and a reduction of white matter, which contains high cholesterol levels, and is linked to decreased cognitive function and insulin resistance [48,49]. In line with this, neuronal insulin resistance reduces cholesterol synthesis and diabetes impairs cholesterol biosynthesis [21]. Conversely, a reduction of cholesterol induces neuronal insulin resistance [22], indicating that a reduction rather than a surplus of cholesterol deteriorates insulin signaling. Our study supports this hypothesis, showing that cholesterol does not exert a negative impact on brain insulin signaling (Figure 4C–F, (Figure 5A,B and Figure 6A,B). It may even be protective against palmitate-induced hypothalamic insulin resistance ex vivo, as CHO + PA was not as detrimental to insulin action as PA treatment on hypothalamic brain slices (Figure 6A,B). Supporting this observation, mice fed a LAR diet were more insulin-resistant compared to STD-fed mice, whereas feeding of CHO + LAR caused lower insulin levels compared to the LAR group (Figure 1B).

Proper neuronal insulin signaling is closely related to mitochondrial homeostasis [6]. In this study, we did not observe altered mitochondrial activity in cells treated with cholesterol (Figure 3B), although cholesterol treatment reduced the expression of mitochondrial genes and mitochondria-regulating genes, such as *Sirt3* and *Pgc1α*, and combined treatment of cholesterol with linoleic acid slightly induced protein carbonylation (Figure 3E,F, Appendix A). So far, it remains unclear why the combined treatment of cholesterol with linoleic acid causes oxidative stress. But, long-term treatment of cholesterol may exert negative effects on mitochondrial activity. It has been shown that reducing cholesterol levels in mitochondria of Niemann-Pick Type C1 mice improves mitochondrial function, suggesting that reducing cholesterol levels in certain conditions can be beneficial for mitochondria [19].

### 4.2. Impact of Fatty Acids on Brain Insulin Resistance

We showed that palmitate is also able to induce insulin and IGF-1 resistance in neurons (Figure 4). Although this observation is not surprising, given the close homology between IR and IGF-1R signaling, it reveals that constant, elevated palmitate concentrations in the brain can be detrimental for brain function, as a reduction of IGF-1R signaling causes massive growth retardation of the brain and induces brain oxidative damage [50,51]. Palmitate induces JNK activation in hypothalamic neurons (Reference [7] and Figure 4) and JNK activation can cause insulin resistance [52], but palmitate-induced JNK activation in neurons does not seem to be a major cause for neuronal insulin resistance (Appendix A). As JNK deficiency in the brain improves insulin sensitivity [53], palmitate presumably induces other kinases in hypothalamic neurons, which are instrumental for palmitate-induced neuronal insulin resistance [7,9]. Interestingly, PA-induced JNK activation was aggravated by the lack of IR (Figure 4B) and IR KO cells showed unaltered insulin sensitivity compared to control cells, presumably due to elevated IGF-1R expression (Figure 4C). To avoid a compensatory upregulation of IGF-1R and to differentiate between IR and IGF-1R signaling (Appendix A), we used the IR inhibitor S961. This experimental design revealed that PA and CHO + PA treatment caused insulin and IGF-1 resistance, highlighting the detrimental impact of the LCSFA palmitate on two key hormones for neuronal homeostasis (Figure 5).

It is still unclear why JNK is increased in IR KO cells. Previous studies have demonstrated that insulin acts as an anti-inflammatory [54,55]. In addition, it has been shown that JNK enhances IR expression in flies [56], suggesting an internal self-regulatory mechanism of neurons to counteract reduced IR expression and action. Although JNK is known to cause serine phosphorylation of IRS1, which is linked to insulin resistance [52], loss of the described JNK-induced Ser307 IRS1 phosphorylation actually deteriorates insulin action and metabolism [57], questioning direct negative effects of JNK activation on insulin action. In line with this, a constitutive active form of JNK overexpressed in hypothalamic Agrp neurons does not impair hypothalamic insulin action [58] and JNK inhibition did not affect PA- and CHO + PA-induced insulin resistance in our study (Appendix A).

It remains uncertain whether LA is good or detrimental for brain function [59]. Most human data originated from epidemiological studies identify associations rather than causations, with contradicting results showing benefits but also risks for metabolism with regards to LA levels [59]. One prospective cohort study revealed that high dietary intake of PUFAs was inversely associated with higher mortality risk, which was mainly driven by LA [60]. LA is incorporated in cardiolipin’s acyl chain, a crucial mitochondrial lipid species [61], and decreased cardiolipin biosynthesis reduces mitochondrial function [62,63]. Therefore, LA seems to be beneficial for mitochondrial function. Yet, LA treatment did not affect mitochondrial respiration in neurons, suggesting that increasing LA levels above normal does not further improve mitochondrial function under basal conditions (Figure 3B). Analyzing mitochondrial respiration with a focus on different substrates and cellular pathways might help to better understand the role of LA in neuronal metabolism. In contrast, it has been previously shown that LA stimulation is sufficient to reduce mitochondrial respiration and increase cell death in hepatoma 1C1 cells [64]. We have observed that combined cholesterol and LA treatment enhances protein carbonylation, suggesting that LA under certain metabolic conditions might be detrimental for neuronal health (Figure 3E–F). Since mitochondria reveal cell- and tissue-specific functions [65], it might explain different results with regards to the influence of LA on cellular homeostasis. We showed that LA in contrast to PA does not induce neuronal insulin resistance (Figure 4A), but in combination with cholesterol can even enhance insulin signaling (Figure 4C). As neurons and glia cells differ in metabolism and interact functionally in vivo, we tested these stimulations on brain slices. We confirmed our in vitro data showing that LA or LA in combination with cholesterol are not detrimental for insulin action in these settings (Figure 6A,B). Yet, further research is clearly needed to investigate whether overall ω6-PUFA levels affect insulin signaling and whether our observation is specific for the ω6-PUFA LA.

### 4.3. Interplay of Cholesterol and Fatty Acids in the Brain and the Periphery

In peripheral tissues, cholesterol can be acquired from endogenous synthesis and from exogenous lipoproteins that deliver sterols from the circulation [47]. Mice challenged with high-cholesterol diets accumulated high levels of cholesterol esters in the liver [5], while only dietary cholesterol in combination with soybean oil-derived fatty acids like the ω6-PUFA linoleic acid in the CHO + SOY diet induced massive oxidative stress, a reduction in mitochondrial protein content and severe inflammation in the liver, demonstrating an interaction of cholesterol and ω6-PUFAs in peripheral tissues [5].

Investigating the brains of mice fed high-cholesterol diets in combination with chow (CHO + STD) or HFDs (CHO + SOY, CHO + LAR) revealed only minor effects on protein carbonylation in the brain (Figure 2E,F) compared to the liver [5], suggesting that the brain is broadly protected by the BBB at a young age. As cholesterol-containing diets reduce the expression of the tight junction protein occludin and reduce the protein expression of some mitochondrial electron transport chain subunits (Figure 1D and Figure 2A,B), it suggests that long-term intake of high-cholesterol diets might harm the brain by increasing BBB permeability. Therefore, our studies (this study and Reference [5]) disclose that tissues react differently to increased dietary intake of fat and cholesterol. In contrast to the brain, the CHO + SOY diet, but not the CHO + STD or CHO + LAR diets, caused hepatic dysfunction [5]. Furthermore, our own unpublished data of this in vivo study revealed that soybean- and lard-based HFDs, independent of cholesterol, impair insulin signaling in muscle, whereas only the LAR diet, but not the high-cholesterol-containing diets, triggered inflammation in white adipose tissue. The liver is able to secrete factors, so-called hepatokines, which are beneficial for metabolism and health. As a stress response, the liver secretes, among others, FGF21 [66,67], which modulates metabolism via acting in the brain [68]. Thus, the mild effect of the different tested diets on brain function might also be explained by regulatory signals from peripheral tissues impacting the brain. Consequently, it is important to assess the effect of dietary components on neuronal and brain function in vitro and ex vivo, to avoid compensatory effects which might mask the consequences of organ-specific nutrient exposure.

Overall, our data have revealed that palmitate and palmitate with cholesterol harm neuronal mitochondrial activity, IR and IGF-1R sensitivity. Importantly, we showed that IR expression is important to mitigate the detrimental effect of PA and CHO + PA on mitochondrial function, insulin signaling and JNK activation. In addition, cholesterol-containing HFDs affect BBB integrity, and cause altered mitochondrial protein expression and metabolic stress in vivo, highlighting the importance of balanced dietary intake of different fatty acids for brain function.

## Figures and Tables

**Figure 1 nutrients-12-01518-f001:**
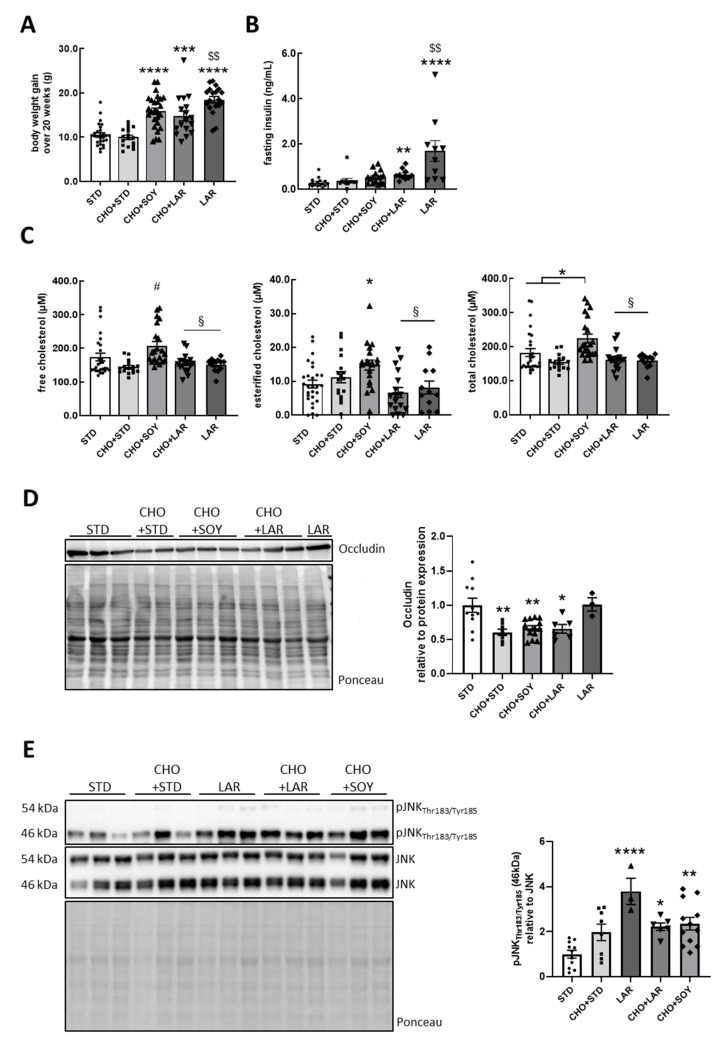
Cholesterol/Poly-unsaturated fatty acids increases cholesterol and metabolic stress in the brain. (**A**) Cumulative body weight change and (**B**) fasting insulin levels measured after a 16 h fast. (**C**) Levels of free, esterified and total cholesterol in hypothalamus. (**D**) Protein expression of tight junction protein occludin and (**E**) of phosphorylated stress kinase c-Jun N-terminal kinase (JNK) Thr183/Tyr185 in the hypothalamus. Dense intensity of occludin was normalized to Ponceau staining and pJNK Thr183/Tyr185 to total JNK protein, which was verified on the same Western blot membrane as a loading control and calculated relative to the standard chow diet (STD) group in each gel. A representative blot is shown. All values are displayed as median ± SEM with a total n of (**A**,**B**) 17–28, (**C**) 12–27, (**D**,**E**) 3–13 mice per group. Statistics: One-way ANOVA with Tukey’s post hoc test for multiple comparisons (**A–E**). * *p* < 0.05, ** *p* < 0.01, *** *p* < 0.001, **** *p* < 0.0001. *: versus STD; #: versus CHO + STD; $$: versus CHO + LAR; §: versus CHO + SOY.

**Figure 2 nutrients-12-01518-f002:**
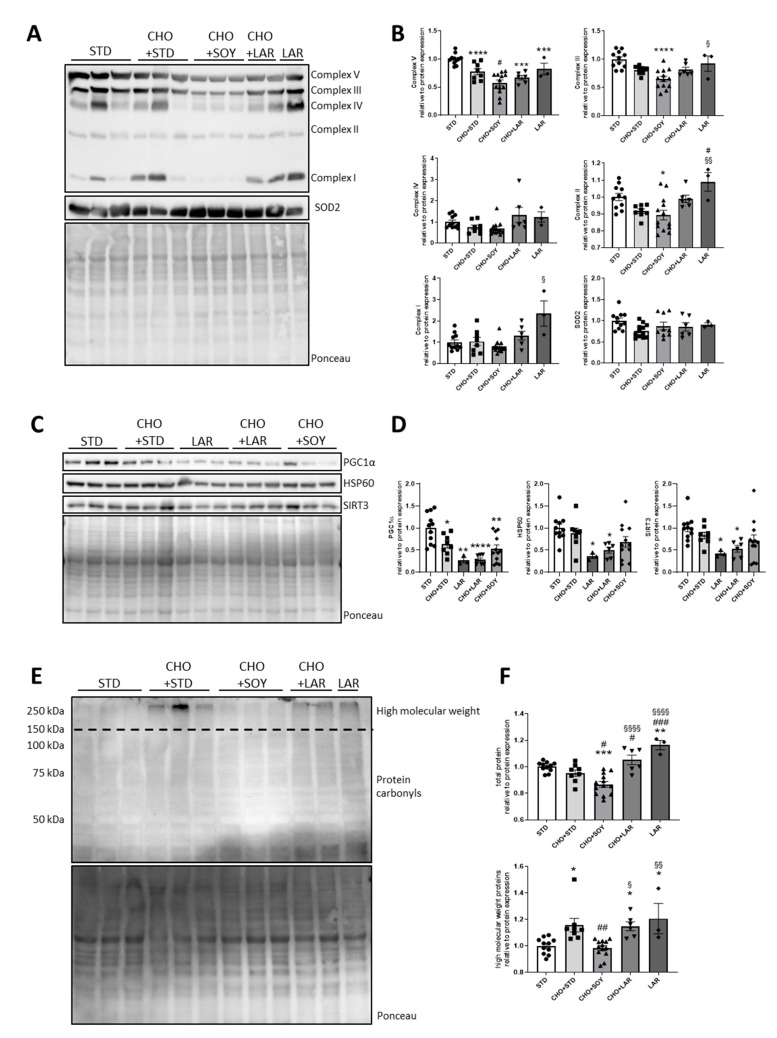
Specific alterations of mitochondrial protein expression due to cholesterol-containing diets. (**A**) Protein expression of subunits of the oxidative phosphorylation complexes (I–V) and SOD2 (Superoxide dismutase 2) in the hypothalamus and (**B**) densitometric analysis. (**C**) Protein expression of PGC1α (Peroxisome proliferator-activated receptor gamma coactivator 1-alpha), SIRT3 (Sirtuin 3) and HSP60 (Heat shock protein 60) and (**D**) densitometric analysis. (**E**) Protein carbonylation as a marker of oxidative stress in the hypothalamus and (**F**) densitometric analysis. Dense intensity was normalized to Ponceau staining, which was verified on the same Western blot membrane as a loading control and calculated relative to the STD group in each gel. Representative blots are shown. All values are displayed as median ± SEM with a total n of 3–13 mice per group. Statistics: One-way ANOVA with Tukey’s post hoc test for multiple comparisons. * *p* < 0.05, ** *p* < 0.01, *** *p* < 0.001, **** *p* < 0.0001. *: versus STD, #, ##, ###: versus CHO + STD and §, §§, §§§§: versus CHO + SOY.

**Figure 3 nutrients-12-01518-f003:**
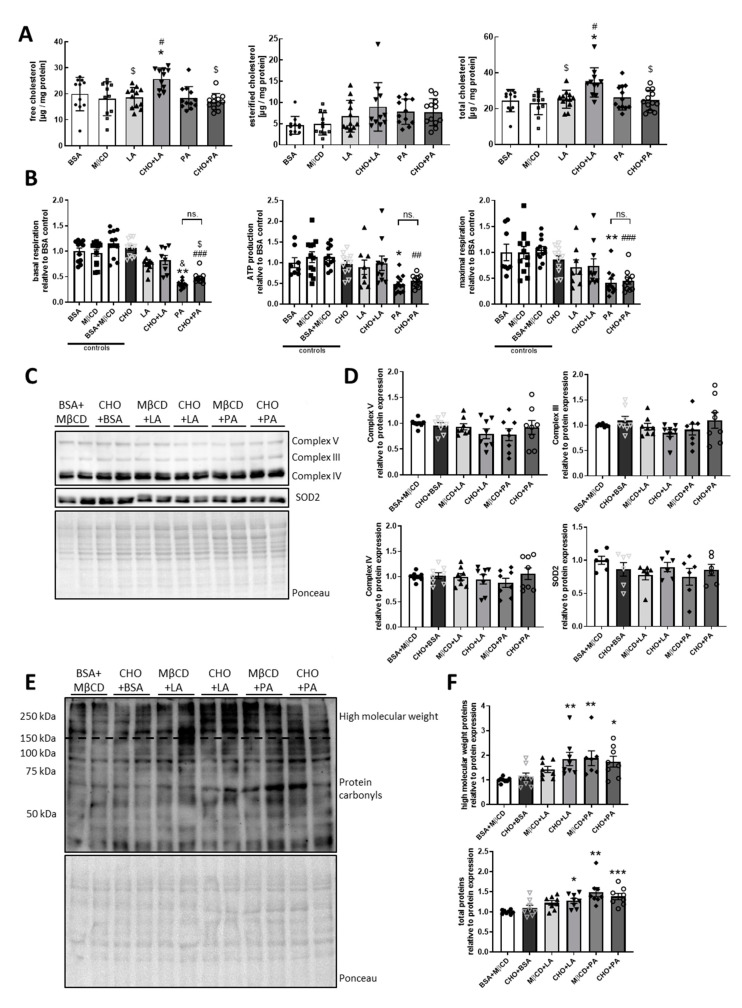
Palmitate, but not cholesterol or linoleate, decreases mitochondrial function. (**A**) Levels of free, esterified and total cholesterol in stimulated CLU183 hypothalamic neurons. (**B**) Relative oxygen consumption rate (basal respiration, ATP production and maximal respiration) in stimulated neurons. A representative experiment is shown. (**C**) Protein expression of subunits of the oxidative phosphorylation complexes (III–V) and SOD2 in stimulated neurons and (**D**) densitometric analysis. (**E**) Protein carbonylation as a marker of oxidative stress in stimulated neurons (high molecular weight proteins > 150 kDa) and (**E**) and (**F**) densitometric analysis. Dense intensity was normalized to Ponceau staining, which was verified on the same Western blot membrane as a loading control and calculated relative to the control (BSA + MβCD) group in each gel. Representative blots are shown. All values are displayed as median ± SEM with a total n of (**A**) 11–12, (**B**) 8–12 and (**C**–**F**) 6–8 per group. Statistics: One-way ANOVA with Tukey’s post hoc test for multiple comparisons. * *p* < 0.05, ** *p* < 0.01, *** *p* < 0.001. Separate depiction of statistics for cholesterol (A) *: versus BSA, #: versus MβCD, $: CHO + LA. Separate depiction of statistics for mitochondrial respiration (B) *: versus BSA, #, ##, ###: versus BSA + MβCD, $: versus CHO + LA, &: versus LA, n.s: not significant. Separate depiction of statistics for (F) *: versus BSA + MβCD.

**Figure 4 nutrients-12-01518-f004:**
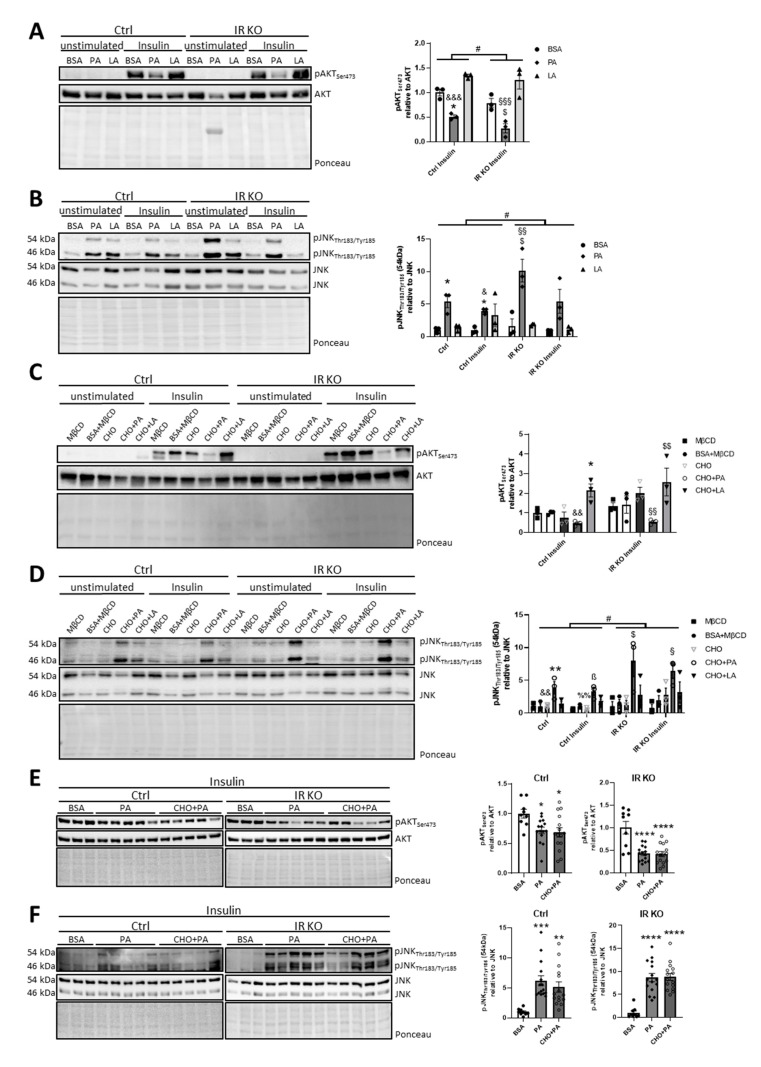
Palmitate, but not cholesterol, induces insulin and IGF-1R resistance with increased inflammation. (**A,C,E**) Protein expression in stimulated control or insulin receptor knockout (IR KO) CLU183 cells of phospho-protein kinase B (pAKT) Ser473, AKT and densitometric analysis. (**B,D,F**) Protein expression in stimulated control (Ctrl) or IR KO CLU183 cells of pJNK Thr183/Tyr185, JNK and densitometric analysis. All values are displayed as median ± SEM. (**A**–**D**) Data of three independent experiments with a total n = 3. (**F**) Data of three independent experiments with a total n = 9–15. Dense intensity of pAKT Ser473 was normalized to total AKT protein, and for pJNK Thr183/Tyr185 to total JNK protein, which was verified on the same Western blot membrane as a loading control and calculated relative to the respective control (BSA or BSA + MβCD) group in each gel. Representative blots are shown. Statistics: Two-way ANOVA with Tukey’s post hoc test for multiple comparisons of Ctrl versus IR KO and one-way ANOVA with Tukey’s post hoc test for multiple comparisons. * *p* < 0.05, ** *p* < 0.01, *** *p* < 0.001, **** *p* < 0.0001. Separate depiction of statistics for A–C *: versus BSA (Ctrl) or BSA + MβCD (Ctrl), &, &&, &&&: versus LA (Ctrl) or CHO + LA (Ctrl), $, $$: versus BSA (IR KO) or BSA + MβCD (IR KO), §§ §§§: versus LA (IR KO) or CHO + LA (IR KO), Separate depiction of statistics for D *: versus BSA + MβCD (Ctrl), &&: versus CHO + PA (Ctrl), ß: versus BSA + MβCD (Ctrl Insulin), %%: versus CHO + PA (Ctrl Insulin), $: versus BSA + MβCD (IR KO), §: versus BSA + MβCD (IR KO Insulin), #: Ctrl versus IR KO. Separate depiction of statistics for **E**–**F** *: versus BSA.

**Figure 5 nutrients-12-01518-f005:**
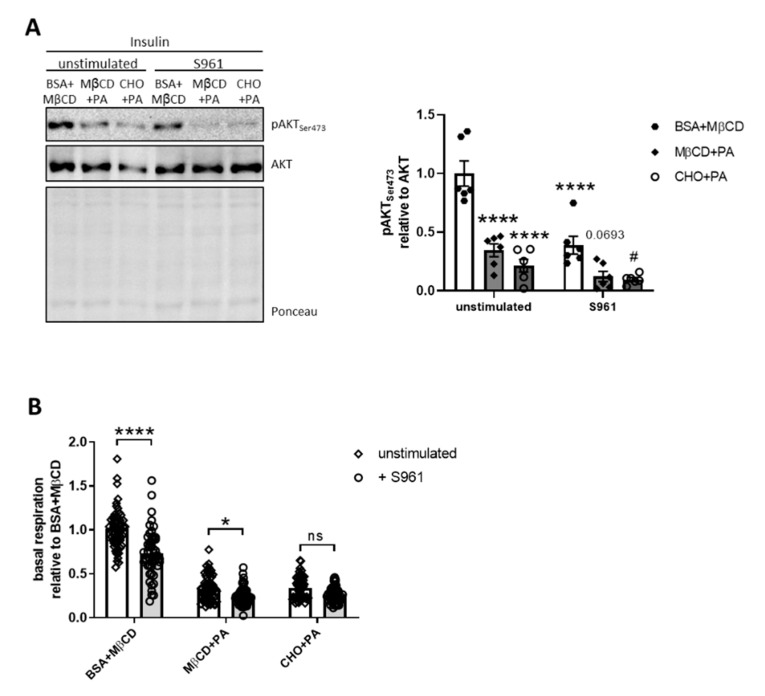
Inhibition of insulin receptor (IR) further increases palmitate-induced IR and insulin-like growth factor-1 receptor resistance. (**A**) Protein expression of pAKT Ser473 and AKT in stimulated neurons and densitometric analysis. (**B**) Relative oxygen consumption rate (basal respiration) in stimulated neurons. Dense intensity was normalized for pAKT Ser473 to total AKT, which was verified on the same Western blot membrane as a loading control and calculated relative to the control (BSA + MβCD) group in each gel. A representative blot is shown. All values are displayed as median ± SEM. (A) Data of three independent experiments with a total n = 6. (B) Pooled data of four independent experiments with a total n = 51–58. Statistics: One-way ANOVA with Tukey’s post hoc test for multiple comparisons., * *p* < 0.05, **** *p* < 0.0001. *: versus unstimulated BSA + MβCD, #: versus S961 BSA + MβCD, 0.0693 versus S961 BSA + MβCD.

**Figure 6 nutrients-12-01518-f006:**
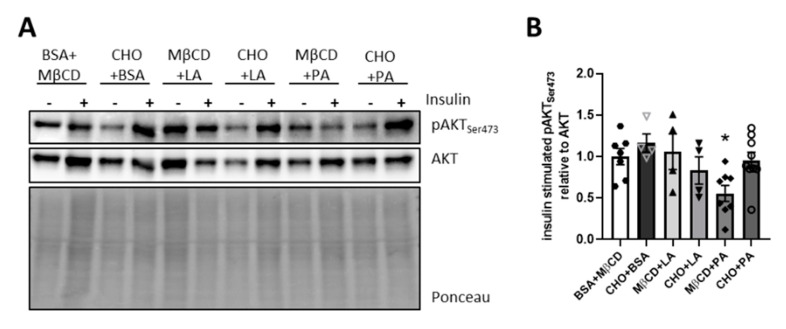
Only palmitate induces insulin resistance in hypothalamic coronal slices. (**A**) Protein expression of pAKT Ser473 and AKT in unstimulated and insulin-stimulated hypothalamic coronal slices and (**B**) densitometric analysis. Dense intensity of insulin-stimulated samples was normalized for pAKT Ser473 to total AKT, which was verified on the same Western blot membrane as a loading control and calculated relative to the control (BSA + MβCD) group in each gel. A representative blot is shown. All values are displayed as median ± SEM. Data of two independent experiments with a total *n* = 4–9. Statistics: One-way ANOVA with Tukey’s post hoc test for multiple comparisons. * *p* < 0.05 versus BSA + MβCD.

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
