# Peer review of "Interplay of Dietary Fatty Acids and Cholesterol Impacts Brain Mitochondria and Insulin Action"

_nutrients, 2020, doi:10.3390/nu12051518_

Round 1

Reviewer 1 Report

This manuscript aims to study how dietary fatty acids and cholesterol, given alone or in combination, impact mitochondrial functions and inflammation within the central nervous system. Via a combination of in vitro and in vivo experiments, the authors demonstrated that the palmitic acid, but not cholesterol, as a negative regulator of insulin signaling and affection inflammation process. The topic explored in this work is very interesting. I have just a couple of minor questions.

  1. In Fig. 1B, the authors showed the fasting insulin levels with different dietary treatment. Please indicate in the figure legends how long they have fast the mice before blood collection.
  2. To demonstrate insulin sensitivity in vivo, the authors only measured serum insulin levels at one time point. It would be much more convincing if they could do a glucose tolerance test or insulin tolerance test in those mice and showed the relevant curves.
  3. Insulin signaling may significantly affect glucose metabolism, e.g. glycolysis, which can be measured by seahorse assay (ECAR measurement). Since they have done seahorse assay, I wonder if the authors can also show ECAR data in the manuscript.

Reviewer 2 Report

The authors examined the impact of obesogenic diets containing either lard or soy oil, with or without high cholesterol content, on hypothalamus insulin resistance, JNK activation and mitochondrial dysfuntion. In addition, in vitro assays were conducted to understand the specific effects of palmitic vs. linoleic acids on the same parameters and on insulin and IGF-1R responses. The experiments were nicely conducted, the manuscript is clearly written, although too wordy given the novelty of the findings. The concepts on which the study was built are quite outdated. It is surprising that the authors did not take into account the new findings in neurodegenerative diseases which involve the trafficking of cholesterol to mitochondria and the dynamics of interaction of mitochondria with other organelles, such as the endoplasmic reticulum. Such an approach would probably have yielded more innovative results. 

Minor comments :

Line 56 : One cannot compare a high fat diet to a ketogenic diet. One causes obesity and insulin resistance due to an excessive intake of calories, the other reduces food intake and therefore induces weight loss with a favourable impact on the response to insulin.

Lines 57-60 : To be accurate, cholesterol aggravates liver steatosis associated with obesity but does not cause obesity.

Fig. 1E and Fig. 2D : please be consistent with the order of the abscissa axis.

Line 616 : PLease, specify in which tissue palmitate induces JNK activation.

Round 2

Reviewer 2 Report

The authors have correctly answered to my comments.